



# IASI Global Radiometric Uncertainty Budget

Dimitrios Kilymis[1], Yannick Kangah[2], Laura Le Barbier[1], Elsa Jacquette[1], Xavier Lenot[3],
Jérémie Ansart[1], Mathilde Faillot[1], Jean-Christophe Calvel[4], Gilles Codou[1], and Olivier Vandermarcq[1]

[1]CNES, Centre Spatial de Toulouse, 18 Avenue Edouard Belin, 31401 Toulouse CEDEX 9, France
[2]SPASCIA, N112 Parc Technologique du Canal, 14 Avenue de l'Europe 31520 Ramonville Saint-Agne, France
[3]CS Group, Zone d'aménagement concerté de la Grande Plaine, 6 Rue Brindejonc des Moulinais, 31500 Toulouse, France
[4]AKKA, 7 Boulevard Henri Ziegler, 31700 Blagnac, France

**Correspondence:** Dimitrios Kilymis (dimitrios.kilymis@cnes.fr)

**Abstract.** The Infrared Atmospheric Sounder Interferometer (IASI) is a Fourier Transform Spectrometer onboard the Eumetsat
MetOp (Meteorological Operational) polar orbit satellite series. The three MetOp satellites (A, B, and C) were launched in
October 2006, September 2012 and November 2018, respectively. IASI-B and IASI-C are still operational, while IASI-A was
decommissioned in November 2021. IASI's mission is to provide accurate atmospheric spectra primarily for meteorological
and climate applications, and as such a high measurement precision is required. Furthermore, the estimation of the measurement
uncertainty is a valuable element, especially concerning climate studies and the establishment of long climate series. This study
presents the global radiometric uncertainty budget as estimated for the IASI instruments. Four major contributors, the correction
of the analog non-linearity, the black body characterization, the scan mirror reflectivity, and the background radiance instability,
have been identified and their impact is presented alongside other minor contributors. For black body or earth view scenes under
investigation, the global budget was found to be lower than 0.2 K for a reference temperature of 280 K, when all uncertainties
were considered fully correlated. These estimates are expected to remain relatively stable throughout the instrument lifetime.

## 1 Introduction

The IASI optical configuration is a variant of the Michelson interferometer measuring thermal infrared radiances (TIR) in the
spectral range 645-2760 cm$^{-1}$ (15.5-3.62 $\mu$m) with an apodized spectral resolution of 0.5 cm$^{-1}$ and a spectral sampling of
0.25 cm$^{-1}$ (Blumstein et al., 2004, 2007). The continuous spectrum is derived from three interferograms in bands situated
between 645-1240 cm$^{-1}$, 1200-2040 cm$^{-1}$, 1960-2760 cm$^{-1}$, respectively. The main operational scientific requirement of
IASI is to provide water vapor and temperature profiles with uncertainties of 10% and 1 K, respectively (Pougatchev et al.,
2009; Bouillon et al., 2022). The large spectral domain of IASI contains several atmospheric constituent absorption bands
allowing the retrieval, among others, of the three major anthropogenic greenhouse gases, namely carbon dioxide (Crevoisier
et al., 2009), methane (Dils et al., 2023) and nitrous oxide (Chalinel et al., 2022).

The IASI lifetime requirement for absolute calibration accuracy is defined as the difference between measured and actual
temperature of a black body at 280 K and equal to 0.5 K. This absolute calibration should be understood as an upper limit
of the bias between observed and real radiances averaged over a large representative ensemble of spectra. Consistently to the



pre-flight performance characterization using on-ground vacuum tests (Blumstein et al., 2004), an on-orbit characterization
performed over IASI-A measurements showed a radiometric accuracy, derived by comparing IASI and the Advanced Very
High Resolution Radiometer (AVHRR) onboard MetOp, well within this specification, having a typical inter-pixel calibration
error of around 0.05 K (Blumstein et al., 2007). Operationally, the in-flight radiometric accuracy of the IASI instruments is
continuously assessed at the IASI Technical Expertise Center (TEC) at CNES by monitoring a series of quality flags and
parameters, as well as regular assessments of the radiometric noise during operation in external calibration mode.
In parallel, a quarterly evaluation of the inter-calibration between operational IASI instruments and with respect to other
high resolution polar orbiting infrared sounders is performed. Currently, inter-calibrations are performed between the op-
erational IASI and CrIS (Cross-track Infrared Sounder) onboard NOAA-20 and NOAA-21 platforms, while in the past the
inter-calibrations also included IASI-A, AIRS (Atmospheric Infrared Sounder) and CrIS, onboard Suomi NPP. These inter-
calibration activities are important since they ensure an additional monitoring of the radiometric and spectral stability of the
instruments. This was recently outlined in the detailed work by Loveless et al. (Loveless et al., 2023), where the excellent
inter-calibration between IASI, CrIS and AIRS was demonstrated.

It is therefore evident that the IASI instrument, thanks to its precision, stability and longevity, has become a key component of
the global climate monitoring, operational meteorology, atmospheric chemistry, as well as numerous other applications (Hilton
et al., 2012; Crevoisier et al., 2013; Clerbaux et al., 2009; Parracho et al., 2021; Joo et al., 2013; Rabier et al., 2009; Vu Van
et al., 2023; Vernier et al., 2024; Wright et al., 2022; Wilson et al., 2024; Guérin et al., 2023). In this scope a quantification of the
uncertainty linked to its acquisitions is important, especially concerning the establishment of long climate series. Additionally,
a characterization of the IASI uncertainty is also required in order to align IASI to the framework of the Global Space-based
Inter-calibration System (GSICS). In recent years similar works have been carried out for other infrared spectrometers, such as
CrIS (Taylor et al., 2023; Tobin et al., 2013) and AIRS (Pagano et al., 2008, 2020), for which a detailed analysis of different
instrumental contributors has allowed to better estimate the radiometric accuracy along the entire spectral range.

The current study addresses the global radiometric uncertainty budget ($U_g$) of the IASI instruments by presenting the contri-
bution of the major uncertainty sources. Since no specification on the global uncertainty exists for the IASI system, a thorough
budget has not been established pre-flight and therefore this work combines elements of pre-flight testing and on-orbit data.
This uncertainty does not take into account the contribution of noise which can be considered as the radiometric sensitivity and
is negligible after averaging, having an impact mostly on single spectra.

The paper is organized as follows: initially, the method for the calculation of the global budget is given in Section 2 and the
contributors taken into account for the establishment of the budget are described in Section 3. The global budget is derived in
Section 4 for both black body and typical earth view scenes and a discussion on the results follows in Section 5.

## 2   Calculation method

The global uncertainty budget $U_g(x,\nu)$ is expressed as the sum of all identified individual contributors $U_c(x,\nu)$ :





**Table 1.** TIGR atmospheric profiles used to simulate earth view spectra. SH and NH stand for South Hemisphere and North Hemisphere, respectively.

| TIGR profile number | Atmosphere type | Surface temperature (K) | Longitude (degree ) | Latitude (degree) |
|---|---|---|---|---|
| 81 | Tropical SH | 298.05 | -140.56 | -18.00 |
| 221 | Tropical NH | 307.92 | 6.61 | 30.27 |
| 681 | Mid-lat NH | 306.82 | 54.02 | 40.68 |
| 1001 | Mid-lat SH | 286.22 | 172.55 | -43.48 |
| 1981 | Polar NH | 243.85 | 161.28 | 68.80 |
| 2181 | Polar SH | 268.45 | -110.32 | -66.15 |

$$U_g(x,\nu) = \sqrt{\sum_{i=1}^{n} U_c(x_i,\nu)^2 + 2 \times \sum_{i=1}^{n-1} \sum_{j=i+1}^{n} r(x_i,x_j) U_c(x_i,\nu) U_c(x_j,\nu)} \qquad (1)$$

where $U_c(x_i,\nu) = ku(x_i,\nu)$ with $u(x_i,\nu)$ the uncertainty due to the parameter $x_i$ at the wavenumber $\nu$ and $k$ the coverage factor (Tansock et al., 2015). A value of $k = 3$ has been chosen, providing a 99.85 % of probability of being lower than the level of $U_c$ when a Gaussian distribution is considered. In Eq. 1 $r(x_i,x_j)$ is the Pearson correlation coefficient between uncertainties from parameters $x_i$ and $x_j$. In the current approach the correlations are considered unknown, and therefore both the favorable and unfavorable cases are taken into account. Their uncertainties are considered either totally independent ($r = 0$) or perfectly correlated ($r = 1$), respectively.

Unless otherwise specified, each uncertainty source is calculated in terms of standard deviation of the difference between a reference spectrum and perturbed spectra. This uncertainty ($U_{rad}(x_i,\nu)$) is then translated into Noise Equivalent Delta Temperature at a temperature equal to 280 K as follows:

$$U_g(x_i,\nu) = \frac{U_{rad}(x_i,\nu)}{\partial Planck(280K,\nu)/\partial T} \qquad (2)$$

The uncertainties have been calculated considering earth view scenes as black bodies at 200 K, 220 K, 240 K, 260 K, 280 K and 300 K. In addition, simulated (daytime) earth view spectra using the 4A/OP radiative transfer model(Scott and Chedin, 1981; Cheruy et al., 1995) and TIGR atmospheric database (Chedin et al., 1985; Achard, 1991; Chevallier et al., 1998) have been used. Their details are given in Table 1.

The calculations presented in this study have been carried out using IASI-B characterization parameters. Their applicability for the other two IASI instruments is discussed in Section 5.





# 3 Contributors

The major contributors to the global radiometric uncertainty budget are presented in this section and are the correction of the

detector non-linearity, the characterization of the internal black body, the scan mirror reflectivity and the instability due to the background radiance. Minor contributors are discussed as well. The global budget does not take into account the contribution of noise, which is considered as random, neither the accuracy of the spectral calibration which could have second order effects. In the current iteration it also does not include the contribution due to the temporal fluctuations of background emissivity.

## 3.1 Detector non-linearity correction

Due to the detector technology, the IASI analog non-linearity only affects the B1 band. It is modeled using a 3rd degree polynomial, which is sufficient in order to linearize the interferograms. However, the residual of the optimization process linked to the definition of the model coefficients is expected to contribute to the global budget.

Given that $I$ and $I_c$ are the measured and corrected raw interferograms, respectively, the corrected interferograms can be written as:

$$I_c = I + A_2 I^2 + A_3 I^3 \tag{3}$$

By separating $I$ into a baseline ($V$) and a modulated ($I_0$) part, we obtain:

$$I_c = I_0 + V + A_2(V^2 + 2VI_0 + I_0^2) + A_3(V^3 + 3V^2I_0 + 3VI_0^2 + I_0^3) \tag{4}$$

The quadratic elements have no impact in B1 and can therefore be neglected. The same goes for the cubic terms whose contribution is also negligible. By centering at the respective baseline, the modulated interferograms can be written as:

$$I_f = I_0 + 2VI_0A_2 + 3V^2I_0A_3 \tag{5}$$

The inverse Fourier transform yields:

$$S_f = S + 2VSA_2 + 3V^2SA_3 \tag{6}$$

where $S$ and $S_f$ are the measured and corrected spectra, respectively.

Therefore, the radiometrically calibrated spectra ($S_r$) can be written as:

$$S_r = \frac{S_{EV}(1 + 2VA_2 + 3V^2A_3)}{S_{BB}(1 + 2V_{BB}A_2 + 3V_{BB}^2A_3)} Planck(T_{BB}) \tag{7}$$

where:





– $S_{EV}$ and $S_{BB}$ are the earth view and calibration black body spectra, respectively. They are considered to be affected by the non-linearity and are corrected using the 3rd degree polynomial.

– $V_{BB}$ are the black body baselines.

– $T_{BB}$ is the internal calibration black body temperature fixed at 293 K.

Given that the lifetime focal plane temperature fluctuations are within 0.2 K for IASI-A and 0.05 K for IASI-B and IASI-C, the analog non-linearity can be considered stable with respect to the temperature. Therefore, the uncertainty contribution lies essentially in the estimation of the coefficients $A_2$ and $A_3$, which have been characterized in flight using verification interferograms. The calculated coefficients, as well as the derived standard errors, for IASI-B are shown in Table 2.

**Table 2.** Non-linearity correction coefficients and corresponding standard errors

| Estimation | $A_2(10^{-2}V^{-1})$ | | | | $A_3(10^{-2}V^{-1})$ | | | |
|---|---|---|---|---|---|---|---|---|
| | PN1 | PN2 | PN3 | PN4 | PN1 | PN2 | PN3 | PN4 |
| Correction coefficients | 2.0251 | 2.1399 | 2.1509 | 2.1349 | 0.0962 | 0.0660 | 0.0817 | 0.1003 |
| Sterr | 0.0037 | 0.0032 | 0.0030 | 0.0023 | 0.0017 | 0.0018 | 0.0021 | 0.0017 |

Therefore, in order to quantify the contribution of the detector non-linearity to the global budget, the operational correction coefficients have been used as a mean of a distribution having a standard deviation equal to the standard errors calculated above. The corresponding calibrated black body and simulated earth view spectra were calculated using equation 7 and their standard deviation was considered as a $1\sigma$ uncertainty due to the non-linearity.

The results for the black body spectra are given in Fig. 1. As expected, the uncertainty is lower for cold spectra and for 110 temperatures closer to the calibration black body temperature. The uncertainties are generally low ($< 1.5 \cdot 10^{-2}$ K), with pixels PN3 and PN4 exhibiting slightly lower values than PN1 and PN2.

In the case of the simulated earth view spectra, given in Fig. 2 and Fig. 3, the calculated uncertainties are lower than those of the black body ones ($< 6 \cdot 10^{-3}$ K), with PN3 and PN4 yielding again slightly lower uncertainties.

## 3.2 Internal Black Body characterization

By fixing the black body emissivity to 1, due to its unknown error, the simplified radiometric calibration is written as:

$$S_r = \frac{S'_{EV}}{Planck(\hat{T_{BB}})}Planck(T_{BB}))\tag{8}$$

where $S'_{EV}$ is the earth view radiance without the background contribution and $\hat{T_{BB}}$, the estimated black body temperature. The uncertainties on this temperature including measurement uncertainty, the non-blackness and the non-uniformity of the black body have been estimated during testing of the PFM detector aboard IASI-B and were found to be equal to 91 mK.




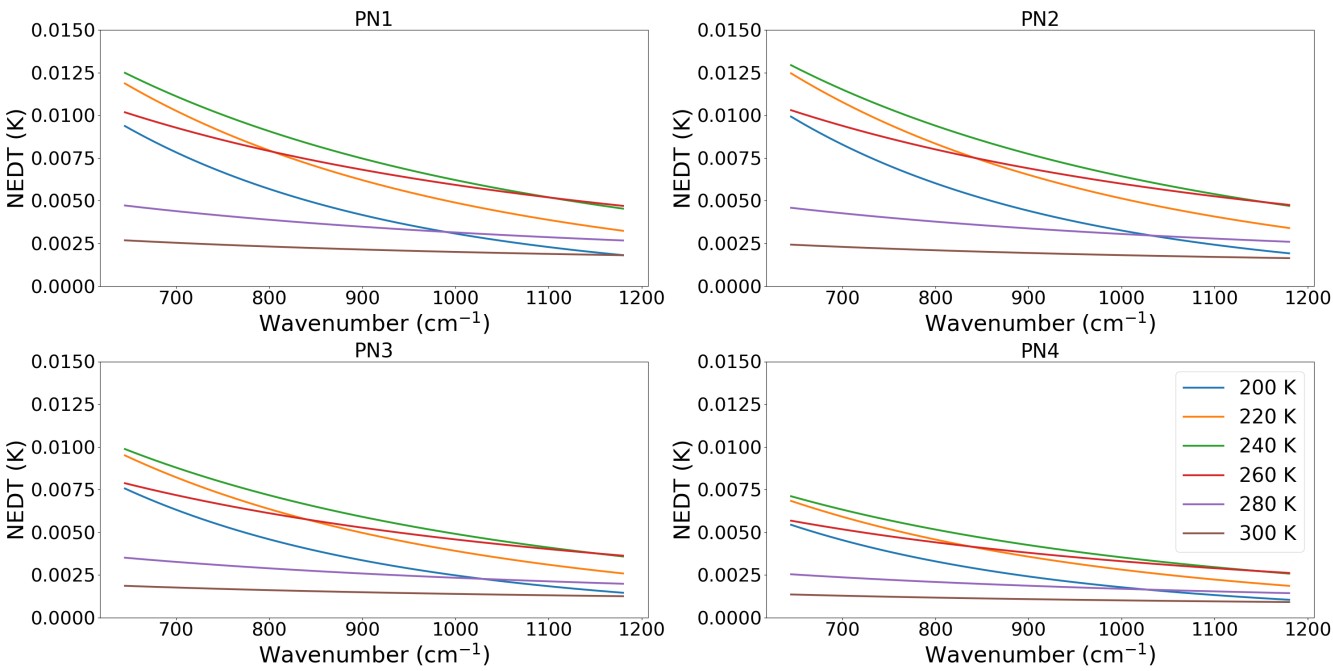

**Figure 1.** Uncertainties due to non-linearity for each pixel calculated on black body spectra at different temperatures.

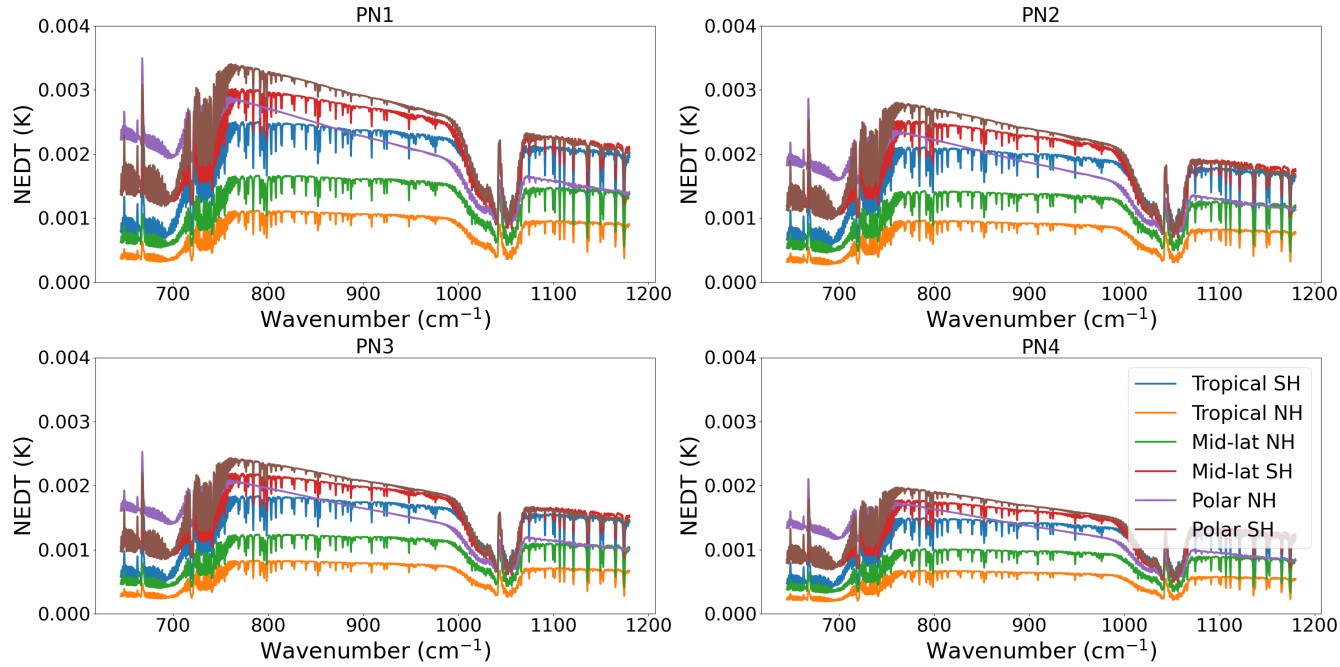

**Figure 2.** Uncertainties due to non-linearity for each pixel calculated on simulated earth view spectra.





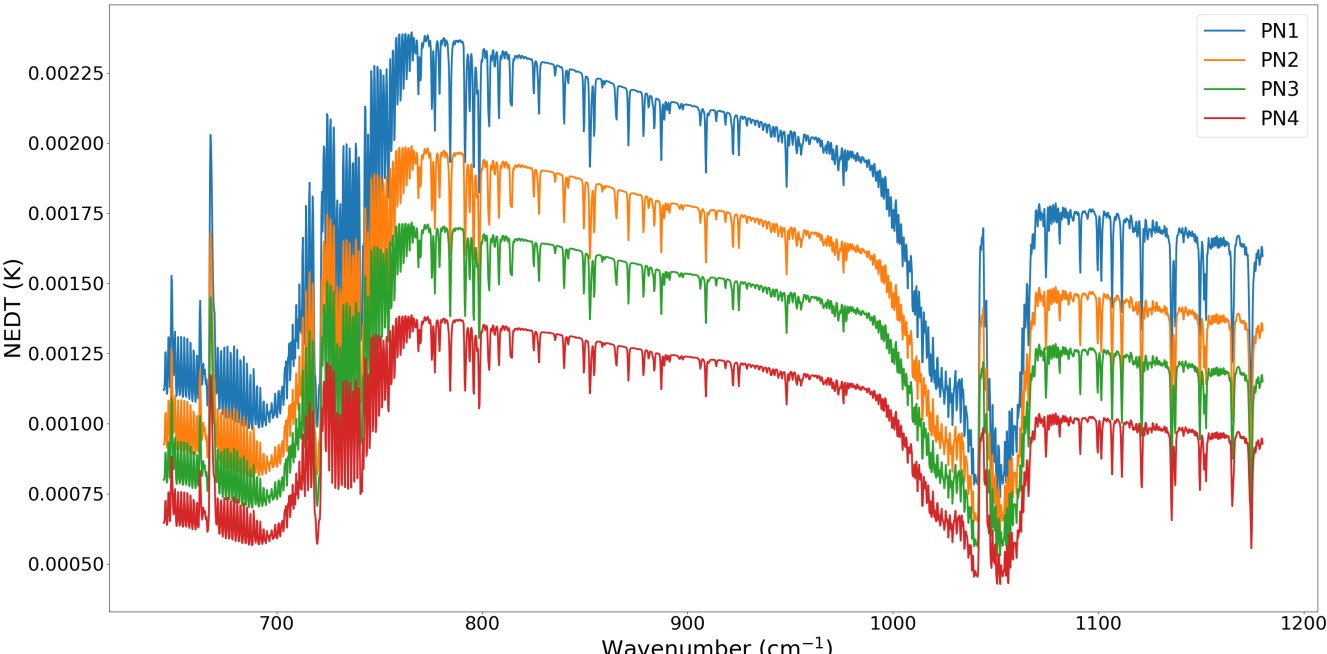

**Figure 3.** Uncertainties due to non-linearity for each pixel calculated on simulated earth view spectra, averaged over the 6 atmospheres.

Therefore, $\hat{T_{BB}}$ has been considered as a normal distribution centered around $T_{BB}$ with a standard deviation of 91 mK. 10,000 black body temperature and calibration slopes have been afterwards drawn and linearly filtered using a filtering length equal to 750 (approximate number of spectra per orbit). The filtered black body temperatures and calibration coefficient slopes were then used to draw synthetic calibration spectra from which $1\sigma$ uncertainties were calculated.

The uncertainties calculated on black body and earth view targets are shown in Figs. 4 and 5, respectively. They generally remain lower than 0.1 K for all targets with warmer black body and surface temperatures yielding the higher uncertainties.

### 3.3 Scan mirror reflectivity

The scan mirror of the IASI instrument is used to select the acquisition target between different scene types: earth view, internal black body, as well as two separate cold space views. In routine operation the scan cycle comprises of 30 earth view scenes having scan angles $\pm 48.33°$ with respect to nadir, then a black body view, followed by a cold space acquisition. The uncertainty on the reflectivity of the scan mirror is mainly due to the temporal variation of the reflectivity law. This law is characterized as a theoretical variation of the reflectivity with the incidence angle on the scan mirror. Its theoretical values are then corrected using the second cold space view (CS2) spectra which actually represent, at first approximation, the reflectivity differences between the first cold space view (CS1) and CS2 viewing incidence angle as follows:





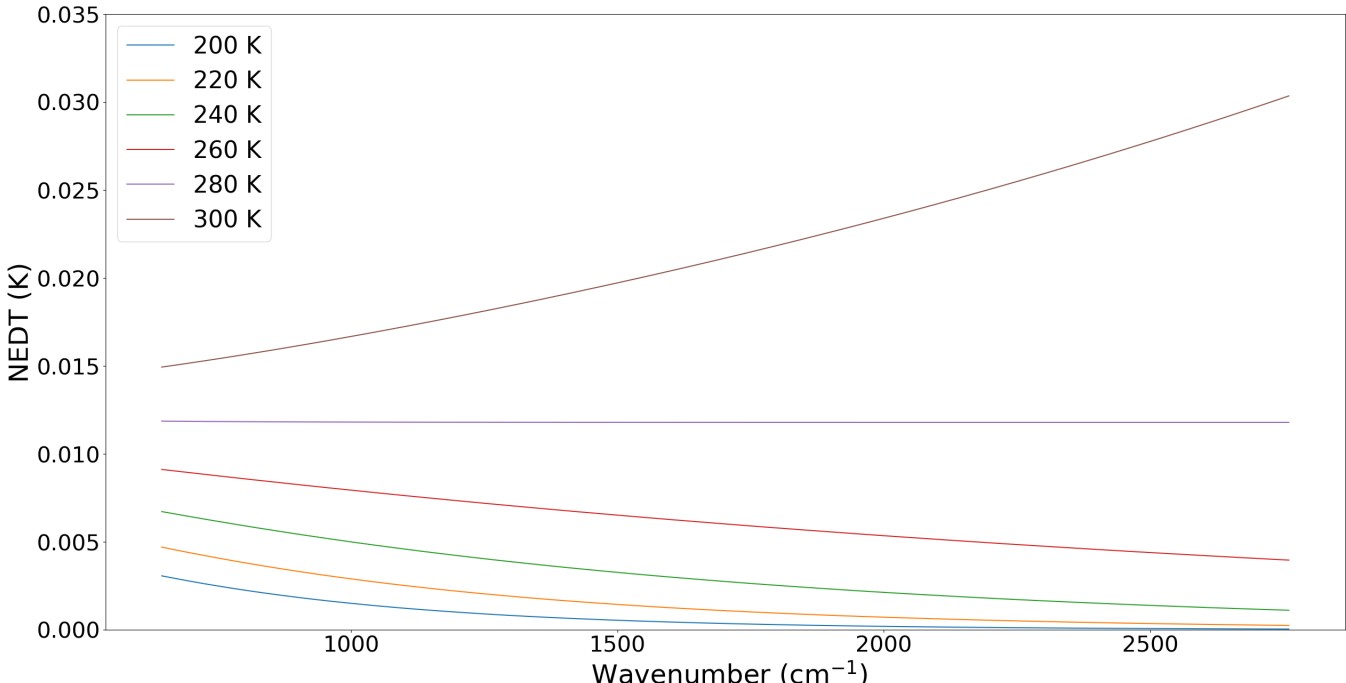

**Figure 4.** Uncertainties due to the calibration black body temperature on black body spectra at different temperatures.

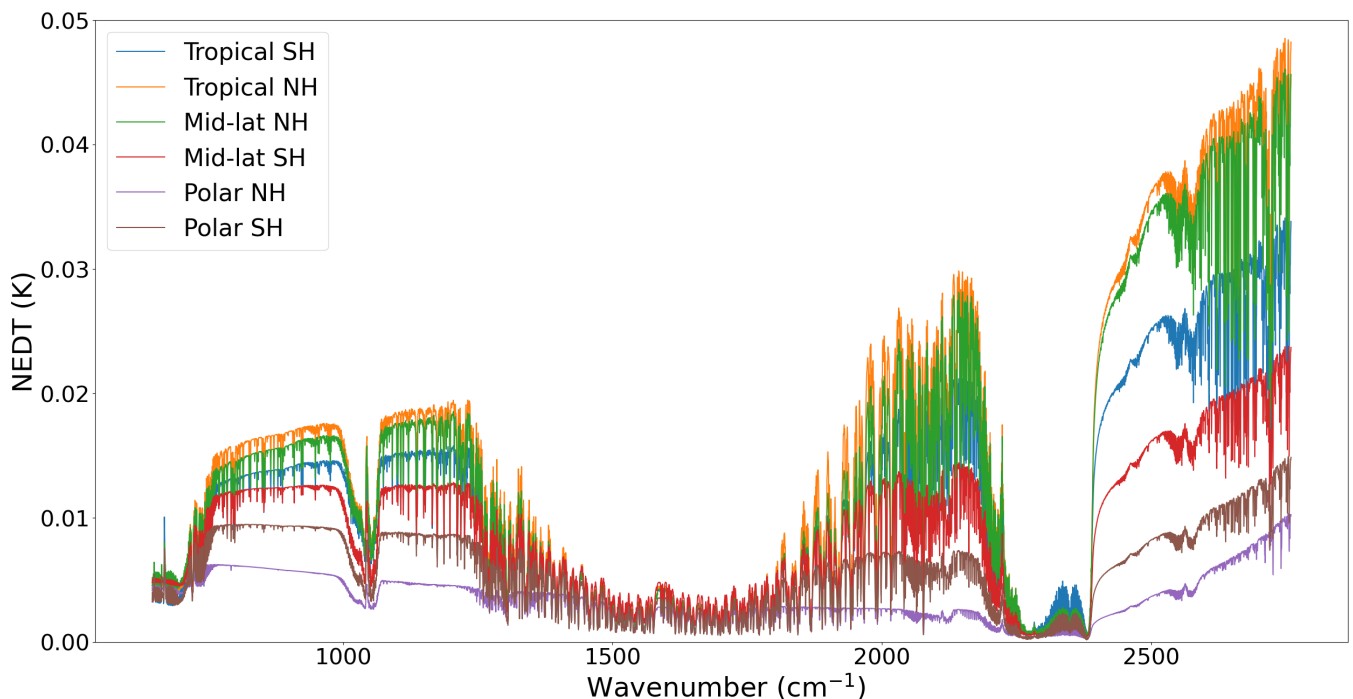

**Figure 5.** Uncertainties due to the calibration black body temperature calculated on simulated earth view spectra.





$$\Delta R_{CS} = R_{CS1} - R_{CS2}(\nu) = \frac{Real\_Part(S_{CS2})(\nu)}{Planck(T_{Scan}, \nu)} \tag{9}$$

where $S_{CS2}$ is the complex spectrum of the CS2 targets and $T_{scan}$ the temperature of the scan mirror. For each scan position (SP), the reflectivity model R(SP,$\nu$) is then corrected as follows:

$$R_{Corrected}(SP, \nu) = R(CS1, \nu) + (R(SP, \nu) - R(CS1, \nu)) \frac{\Delta R_{CS}}{R(CS1, \nu) - R(CS2, \nu)} \tag{10}$$

The reflectivity model correction described by equation 10 is not applied rigorously. The radiometric post calibration correction due to the angular variation of the scan mirror reflectivity is applied by using the knowledge of the variation model: the

radiometric accuracy of the calibrating spectra using the raw model and the corrected one are monitored by IASI TEC and the the reflectivity variation law is updated when this radiometric error reaches 0.1 K. Figure 6 shows typical values of the maximal radiometric errors due to the scan mirror reflectivity law for different target temperatures, right before the model update. These errors will be hereafter considered as proxies of the $3\sigma$ uncertainties due to scan mirror reflectivity. This way, the worst case scenario is considered for the corresponding contribution to the global budget.

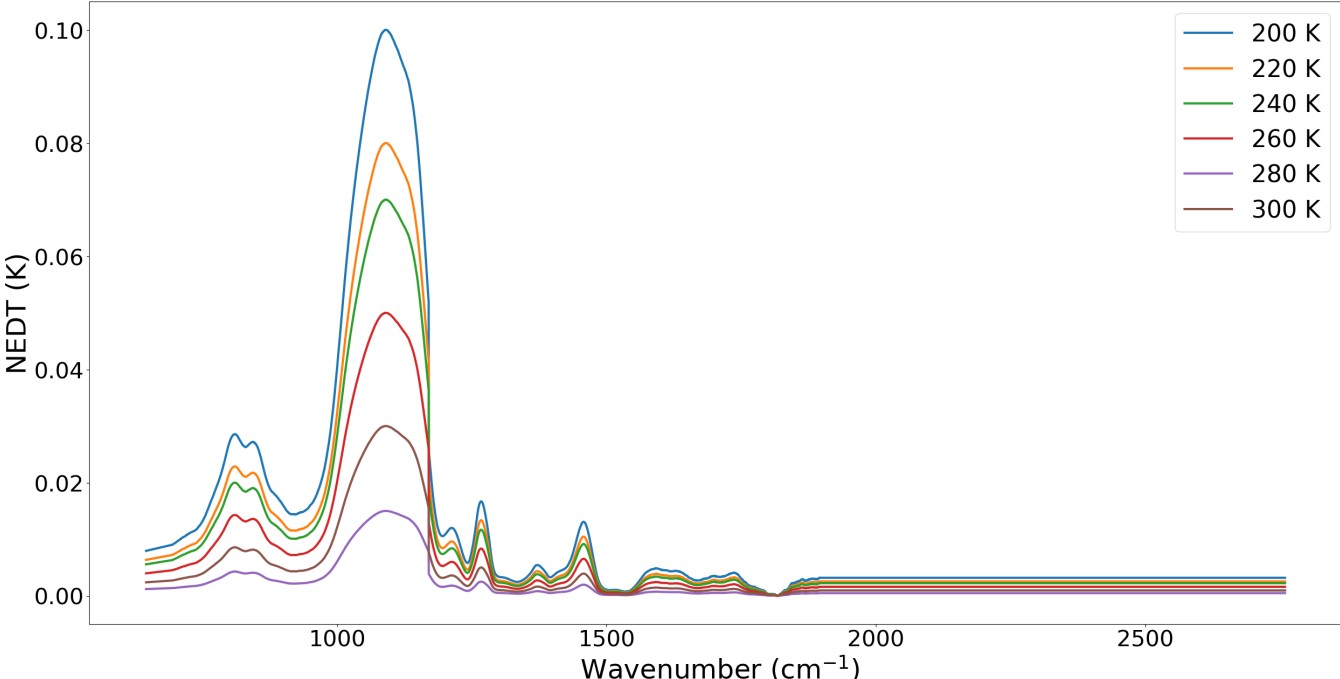

**Figure 6.** Maximal radiometric errors due to scan mirror reflectivity uncertainties.




## 3.4 Background radiance instability

A possible instability of the temperature of different instrument components during the calibration process may affect the intensity of Earth View and Cold Space straylight. An evaluation of the impact has been performed by carrying out pessimistic assumptions, since the uncertainties linked to each individual temperature could not be perfectly established. The study concluded that the only instrument component whose temperature instability could potentially impact the radiometric calibration is the Interferometer and Hot Optics subsystem (IHOS) with the corresponding calibration error being around 0.01 K on the whole spectrum, except for wavenumbers lower than 700 cm $^{-1}$ where it reaches up to 0.02 K. This contribution has been taken into account using a pessimistic scenario with a linear model from 0.02 K at the start of B1 to 0.01 K at 700 cm $^{-1}$ and a stable contribution up to the end of B3.

## 3.5 Minor contributors

In parallel to the major contributors presented above, several minor ones have been identified but not taken into account in the establishment of the global budget due to their relatively low contribution. These notably include:

- **Scan mirror temperature instability:** The correction of the scan mirror reflectivity takes into account the temperature of the back of the mirror, its contribution to the global budget is therefore linked to the accuracy of the measurement. An estimation of this contribution can be deduced using the fluctuations in the measurement of the temperature of the back of the scan mirror, performed at each external calibration. In the case of IASI-B the fluctuations with respect to the average temperature have been measured between $2.5 \cdot 10^{-2}$ K and $9 \cdot 10^{-2}$ K, therefore having a negligible impact on the radiometric calibration.

- **Scan angle characterization error:** Ground testing has shown that the measurement error for the scan angle is lower than 1 mrad. This corresponds to a negligible radiometric error in the order of $10^{-6}$ K.

- **Interferogram phase instability:** The only phase dependent term taken into account in the calibration law is the instrument transmission factor. As the phase is designed to be sufficiently stable, this theoretical uncertainty has no effect on radiometry.

## 4 Global uncertainty budget

The global budget computed by combining all the estimated uncertainties derived from the major contributors on simulated black body and averaged earth view targets using Eq. 1 is given in Fig. 7 and Fig. 8, respectively.

Qualitatively, the signatures of the major contributors are similar for both target types. The budget is quasi-uniformly influenced by the contributions of the uncertainty in the black body characterization and the background radiance instability with the exception of the start of B1 where the latter gradually increases. B1 is also dominated by the signature of the contribution of the scan mirror reflectivity, especially between 1000-1200 cm $^{-1}$. On the other hand the contribution in B1 due to the correction





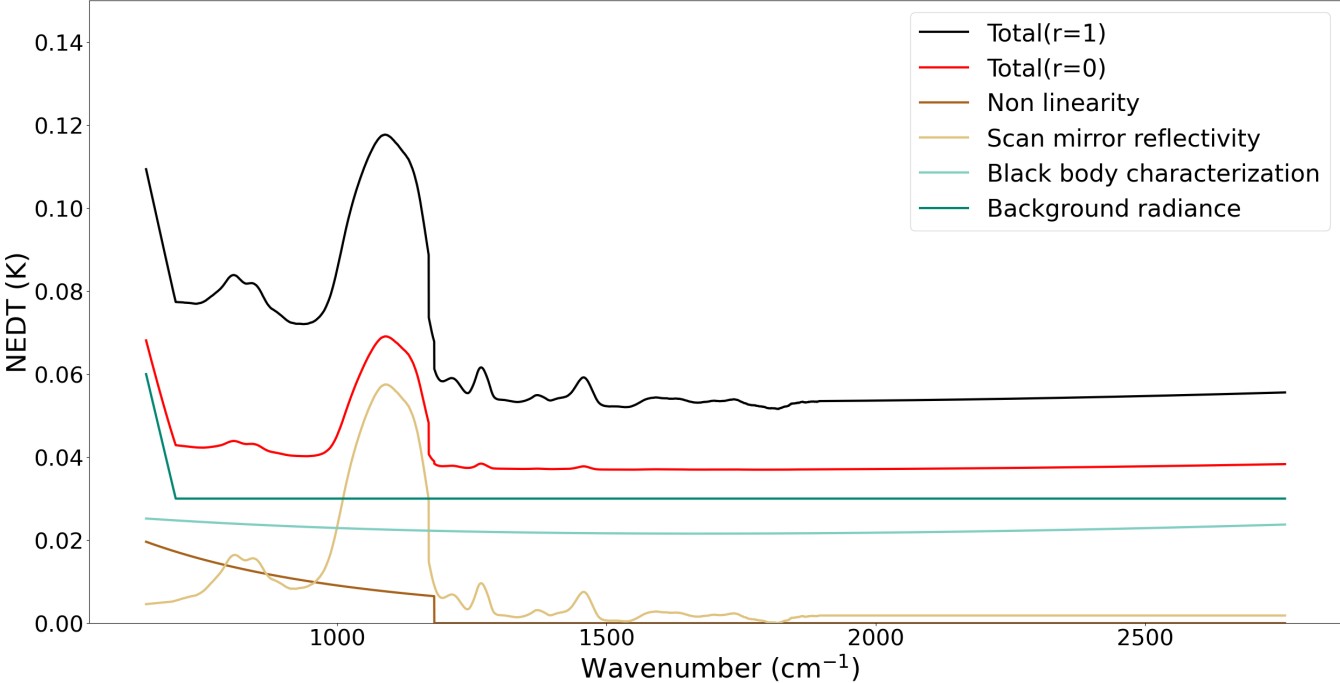

**Figure 7.** $U_g$ calculated over black body scenes considering that all uncertainties are perfectly correlated (r=1) and fully independent (r=0), alongside the major contributors.

of the analog non-linearity is relatively weak. It is worth noting that for earth view scenes the global uncertainties towards the end of B3 (driven by the contribution of the black body characterization) are comparable, if not more important, than the ones around the contribution from the scan mirror reflectivity around 1000-1200 cm$^{-1}$.

In the case of black body targets, the maximum of the global uncertainty budget is located around 1100 cm$^{-1}$ and is equal to 0.115 K and 0.068 K in the fully correlated and uncorrelated cases, respectively. For earth view scenes the maximum is also located around 1100 cm$^{-1}$ in the fully correlated case with a value of 0.127 K, whereas for the uncorrelated uncertainties it is found at the end of B3 with a value of 0.080 K.

Concerning the dependence on target type or temperature, the results for the black body are given in Fig. 9. For both correlated and uncorrelated cases, the global uncertainty increases, driven by the contribution due to the uncertainty of the black body characterization. The only exception is the region around 1000-1200 cm$^{-1}$ in which lower scene temperatures produce relatively higher uncertainties. For the earth view scenes, shown in Fig. 10, there also exists a dependence on the scene type but even for warm surface temperatures (tropical NH and mid-latitude NH), the uncertainties are in the worst case equal to 0.165 K.



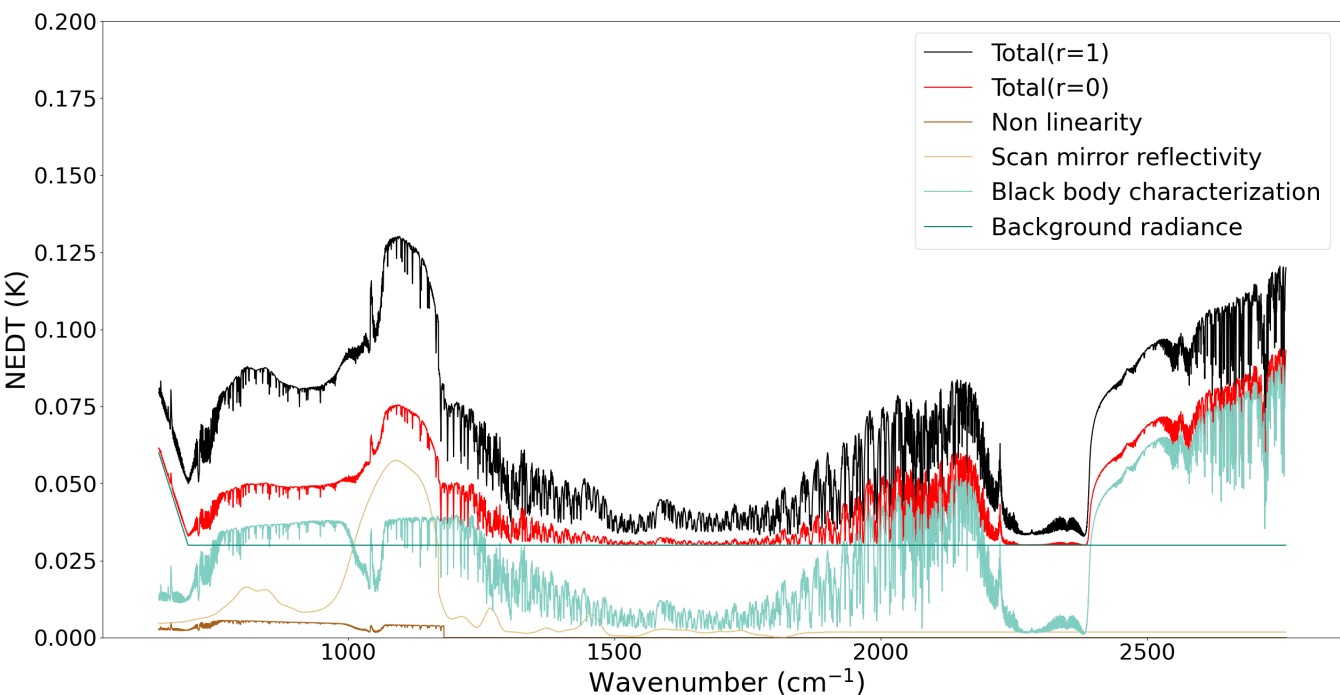

**Figure 8.** $U_g$ calculated over averaged earth view scenes considering that all uncertainties are perfectly correlated (r=1) and fully independent (r=0), alongside the major contributors.

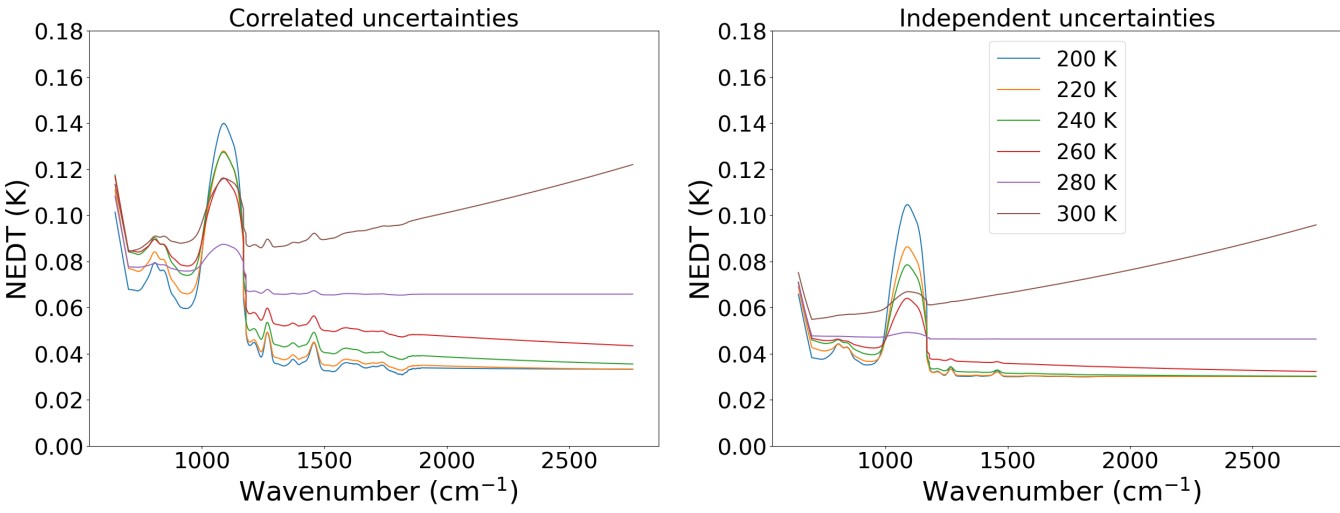

**Figure 9.** $U_g$ calculated over black body scenes considering that all uncertainties are perfectly correlated (left) and fully independent (right).





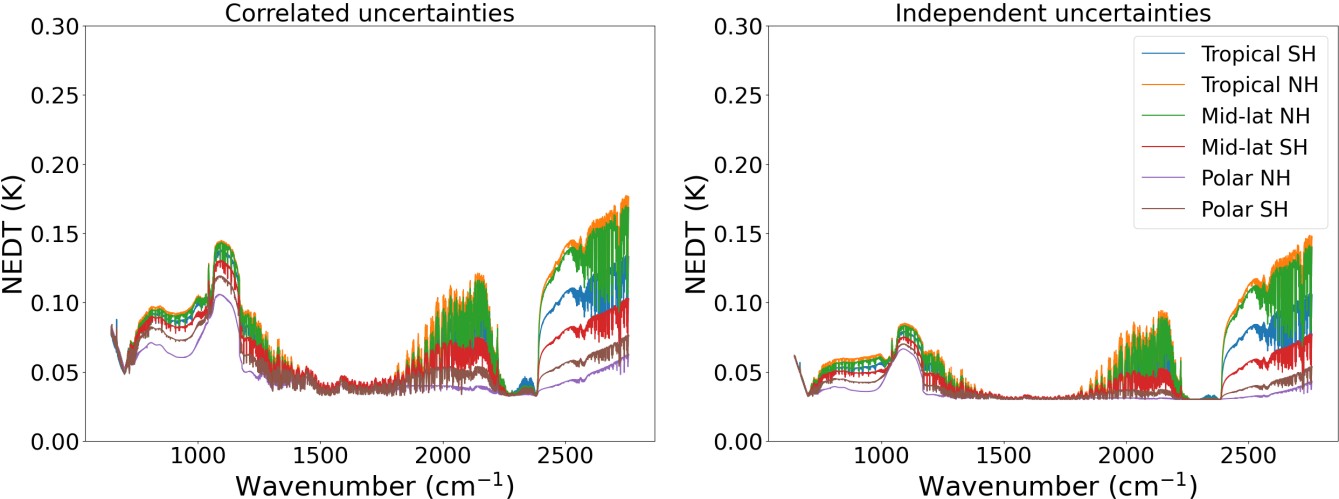

**Figure 10.** $U_g$ calculated over earth view scenes considering that all uncertainties are perfectly correlated (left) and fully independent (right).

## 5 Discussion

A different, albeit less precise, approach for the estimation of the global radiometric uncertainty, as well as its stability, is via

comparisons against referenced data or other well characterized and calibrated measurements. At the IASI TEC, the Sounders Inter-Calibration (SIC) software is routinely used in order to perform inter-comparisons for stability monitoring between IASI instruments, as well as IASI and other TIR sounders. These inter-comparisons are performed by defining overlapping soundings (called hereafter CNO for Common Nadir Observations) from consecutive orbits and selecting the most radiometrically stable and homogeneous scenes. Fig. 11 shows a typical annual comparison between IASI-B and IASI-C for a selection of CNO.

The averaged bias is less than 0.1 K which is consistent to the global uncertainty estimation. In addition, this difference is very stable in time within 0.1 K, as shown in Fig. 12 and in comparison to earlier results (Le Barbier et al., 2021). The standard deviation is higher (roughly around 0.5K), since single CNO are more impacted by time and space coregistration errors and radiometric noise.

The SIC software is also currently used to compare IASI-B/C spectra against those obtained by CrIS on board NOAA-20

and NOAA-21. For that purpose IASI CrIS-Like spectra (i.e IASI at CrIS spectral sampling) are obtained by replacing the IASI apodisation function by the one of CrIS. Fig. 13 and Fig. 14 give typical differences between IASI-B and CrIS on board NOAA-20 and NOAA-21, respectively. In all cases the maximal biases are lower than 0.15 K and not significantly evolving over time, due to the remarkable stability of both IASI and CrIS. This value is close to the results recently reported by Loveless at al. (Loveless et al., 2023) who performed intercomparisons between IASI and CrIS SNPP and NOAA-20 and found the

maximal biases to be lower than 0.25 K. Their approach also reproduces the feature in the $O_3$ absorption band (around 1000-1100 cm$^{-1}$) which could be linked to the contribution stemming by the scan mirror reflectivity characterization.



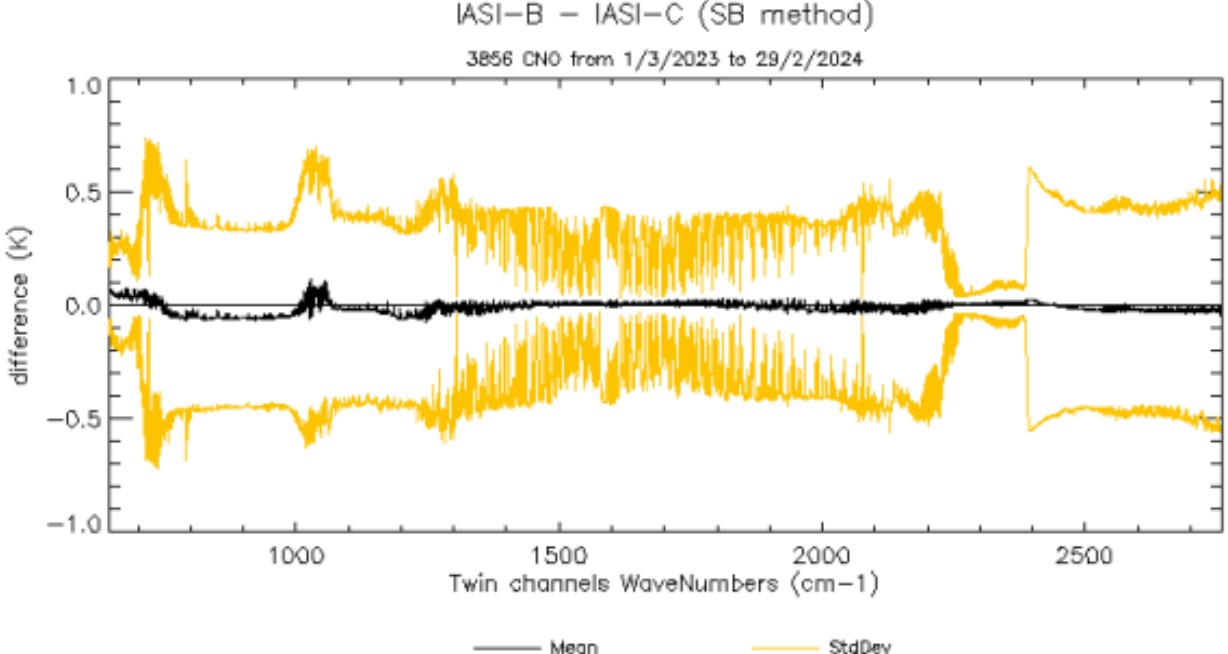

**Figure 11.** Mean NEΔT at 280K radiometric bias (black curve) and standard deviation with respect to the mean (yellow, one per instrument) of an annual IASI-B – IASI-C inter-comparison.

The feature in the sharp $O_3$ rays could also be partially linked to residual spectral shifts. In the IASI system monitoring and requirements the spectral performances, and therefore their contributions to the uncertainty budget, are considered separately from the radiometric ones, since the former ones are evaluated in NEΔT and the latter in terms of maximum relative spectral

shift. The corresponding specification of $\delta\nu/\nu < 2 \cdot 10^{-6}$ for all channels is largely met and particularly stable for all IASI instruments. As such, a possible contribution arising from the IASI spectral response characterization is not expected to have a major effect and has not been explicitly considered in the current study but should be included in a future iteration of the uncertainty budget for the sake of completeness.

     An important question regarding the global uncertainty budget is its evolution over time, since the lifetime of IASI-A was

extended to fifteen years, and it is expected that IASI-B and IASI-C will reach comparable operational lifetimes. In this sense a precise calculation of the evolution has not been attempted in the current work. However it is possible to give a rough estimation of the evolution of each major contributor. the accuracy of the non-linearity correction, and therefore the residual bias, is linked to the temperature of the focal plane, whose lifetime evolution is equal to 0.2 K for IASI-A and 0.05 K for both IASI-B and IASI-C. Given the relatively low contribution of the non-linearity correction to the global budget, it is not expected

that the above evolution will have a significant effect. The evolution of the optical bench temperature is also expected to affect the lifetime stability of the global uncertainty, through contributions to the detector non-linearity, as well as the background



**Figure 12.** Temporal evolution of IASI-B-IASI-C NEΔT at 280K radiometric bias integrated over three pseudo-bands.





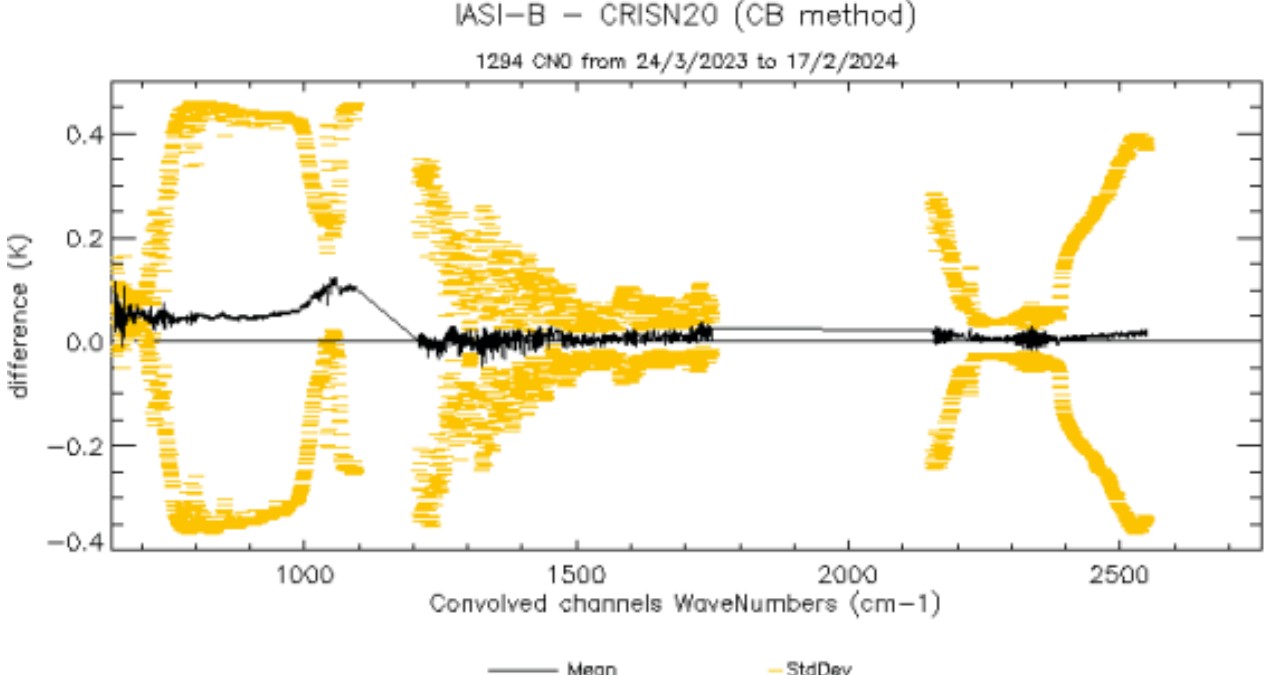

**Figure 13.** Mean NEΔT at 280K radiometric bias (black curve) and standard deviation with respect to the mean (yellow, one per instrument) of an annual IASI-B – CrIS on board NOAA-20 inter-comparison.

radiance instability. For all three IASI instruments the optical bench temperature variation has been within 0.05 K during the instrument lifetime and therefore negligible.

As for the contribution from the long term temporal evolution of the scan mirror reflectivity, each update of the model coefficients takes into account the ageing and degradation of the scan mirror surface. The evolution of the uncertainty could be approximated by comparing the residuals after each update, however the statistics are not adequate in order to draw a firm conclusion. Nonetheless, the available data does not show an evident evolution of the correction residuals with time.

Finally, although the study was carried out using IASI-B data, the results are transferable at a first approach to the other two IASI instruments as well. The contributors that could potentially have a notable impact on the global budget are the black body
characterization and the background radiance instability, since for the scan mirror reflectivity a similar assumption can be made as in the case of IASI-B. Concerning the black body characterization, the corresponding contribution is slightly lower for IASI-A and IASI-C, since the estimated temperature measurement uncertainties are 7 mK lower than for IASI-B. In the case of both black body and earth view global budget, this discrepancy results to a 3 mK and 1 mK decrease around $1100 \, \text{cm}^{-1}$ for perfectly correlated and uncorrelated contributions, respectively. As for the background radiance instability, the IHOS temperature for
all three IASI is within 0.1 K in their lifetime, therefore no difference is expected for IASI-A and IASI-C.





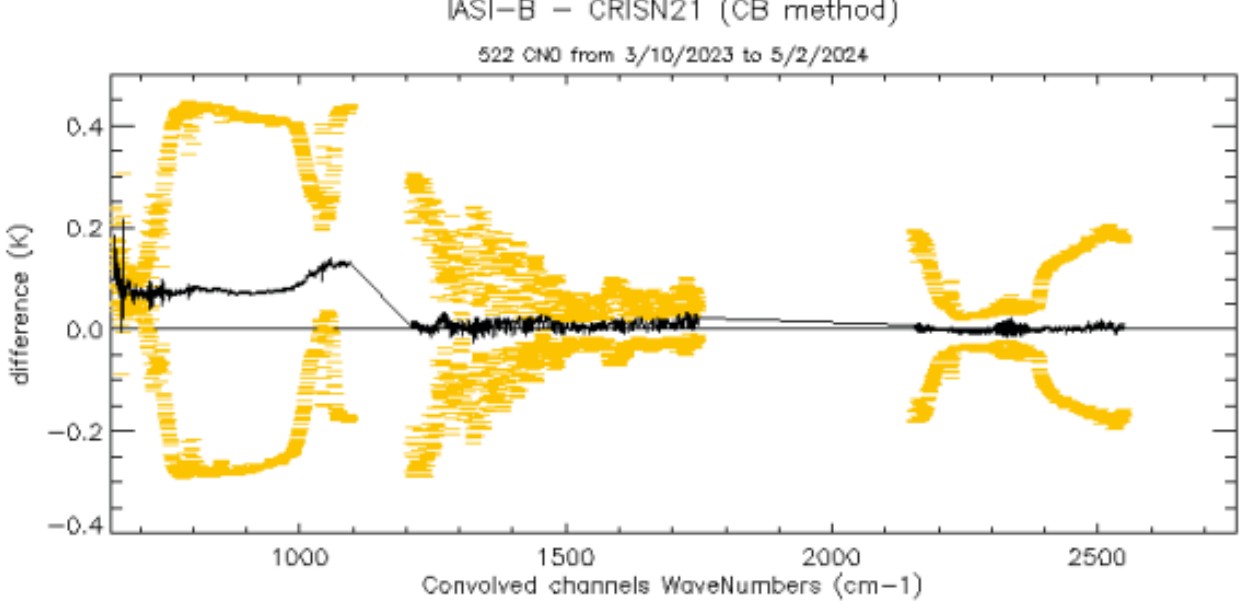

**Figure 14.** Mean NEΔT at 280K radiometric bias (black curve) and standard deviation with respect to the mean (yellow, one per instrument) of an IASI-B – CrIS on board NOAA-20 inter-comparison over four months.

## 6 Conclusions

The IASI global radiometric uncertainty budget has been estimated on IASI-B data, where available, by combining the individual radiometric uncertainties from the major contributors : the correction of the analog non-linearity, the black body characterization, the scan mirror reflectivity and the background radiance instability. For the first two, the radiometric contributions have been evaluated using theoretical radiometric transfer functions together with in-flight parameters. For the scan mirror reflectivity characterization the maximal allowed radiometric error threshold has been used as the $3\sigma$ uncertainty, whereas in the case of the background radiance instability a pessimistic estimation was also taken into account. Two scenarios were considered, one where the above contributions are completely correlated and one where they are uncorrelated. The global radiometric uncertainty budget was found to be well below 0.2 K for all black body and typical earth view scenes considered in the fully correlated case, whereas it is slightly lower in the uncorrelated one. The black body characterization and background radiance have a significant and even contribution over the entire spectral range, whereas the scan mirror reflectivity gives a notable signature in B1. The global uncertainty is expected to be similar for the three IASI instruments and not evolve significantly over the instrument lifetime.




*Author contributions.* DK, YK, LLB, EJ and XL performed the study over the past years. JA provided feedback on the instrument. MF and JCC have provided necessary data. DK and YK prepared the manuscript. GC and OV contributed remarks and revisions on the manuscript.

*Competing interests.* The authors declare that there are no competing interests.

*Acknowledgements.* The authors would like to thank Carsten Standfuss for the insightful comments on this work.



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
