# Peer review of "IASI Global Radiometric Uncertainty Budget"

_EGUsphere, 2025_

## Author Response (AR1)

Dear Dr. Hoffmann,

We hereby submit the revised version of our manuscript, which responds to the reviewers' comments. Following are the responses to the comments, as posted in the preprint Discussion section.

Sincerely, Dimitrios Kilymis

**• Reviewer 1:**

This study provides a well-structured analysis of the global radiometric uncertainty budget for the IASI instruments, highlighting the main contributors such as analog non-linearity correction, blackbody characterization, scan mirror reflectivity, and background radiance instability. The results demonstrate that the overall uncertainty remains below 0.2 K at 280 K, ensuring long-term stability. Given the quality and relevance of this work—particularly in view of upcoming missions like IRS, IASI-NG, HIRS, and GIIRS—I strongly recommend this paper for publication.

**Response**

We would like to thank the reviewer for the positive feedback.

**• Reviewer 2 and authors' response:**

This paper presents the radiometric uncertainty budget of the IASI instruments by taking the leading error sources into account and applying a rigourous error propagation. The paper is clearly written and presents the estimated error for various scenes.

We would like to thank the reviewer for the overall appreciation of the manuscript.

**General comment:**

Although the impact of the various error sources is well elaborated, I am missing some details about the error sources themselves. I would like to learn more about the estimates of these errors or at least find a reference where more details about these estimates are given (see corresponding comments below).

In this manuscript we have tried including as much information derived from internal documentation as possible. However, in some limited cases the relevant references are documents with restricted access and we are therefore unable to provide the full details. In the revised version of the text we have added some additional context that will hopefully make the manuscript more thorough.

**Comments:**

**Line 88:**

"The quadratic elements have no impact in B1 and can therefore be neglected. The same goes for the cubic terms whose contribution is also negligible."

I assume they do not contribute to the spectrum because they are spectrally out of band? If so, I would describe it like this. If not, please explain why the contribution is negligible.

The quadratic elements are indeed out of band in the spectral space. The cubic term has been shown to be a negligible dispersion component which can therefore be considered constant for a given scene temperature [EUM/OPS/DOC/23/1345895]. Relevant comments and reference document have been added in the manuscript.

**Line 99:**

"Given that the lifetime focal plane temperature fluctuations are within 0.2 K for IASI-A and 0.05 K for IASI-B and IASI-C, the analog non-linearity can be considered stable with respect to the temperature."

What about detector aging? Could this change the non-linearity over the lifetime? (Same question holds for line 220.)

Detector ageing could indeed be considered in principle as a factor affecting the performance of the non-linearity correction. However, the studies performed during the development of the non-linearity correction algorithm showed that its impact should be marginal, as long as the number of interferograms taken into account is sufficiently large in order to cancel out the increased noise. We have added a relevant comment in the revised manuscript.

**Line 102:**

"Therefore, the uncertainty contribution lies essentially in the estimation of the coefficients A2 and A3, which have been characterized in flight using verification interferograms."

Here I would like to have some more information on what these "verification interferograms" are, how the coefficients have been characterised and where the standard errors come from. These standard errors seem to originate from a statistical determination (e.g., using several verification interferograms). Can you rule out systematic biases which are not reflected in these standard errors?

Due to bandwidth limitations, the IASI instruments do not routinely transmit interferograms to the ground. However, for each scan line, a single interferogram for one pixel and one band is transmitted, in the so-called Verification Data. The pixel and band for which the verification interferograms are going to be transmitted can be programmed in advance. In this context, a collection of black body verification interferograms is acquired for the characterization of the nonlinearity correction. The corresponding spectra are then treated in order to remove the non-linearity artefacts appearing outside the useful band and the correction polynomial parameters Ax are calculated using an iterative scheme.

We have revised the text in order to provide the above information.

Addressing the issue of systematic biases inherently involves assessing the accuracy of the Non-linearity correction method. This is intrinsically linked to the accuracy of the determination of various measurement parameters—such as the instrumental phase, the constant measurement offset, and the spectral responsivity. This accuracy has been thoroughly investigated on ground by the manufacturer through simulations of the measurement process and Level 1 data processing. These studies have demonstrated that the dominant source of biases—namely, the accuracy of baseline estimation—induces negligible systematic deviations, typically on the order of centikelvin. However, absolute systematic errors due to the Non-linearity correction, which are by definition not measurable in flight, fall outside the scope of the present study, since it focuses on the uncertainty component associated with the precision in determining the non-linearity correction polynomial.

**Line 117:**

"The uncertainties on this temperature including measurement uncertainty, the non-blackness and the non-uniformity of the black body have been estimated during testing of the PFM detector aboard IASI-B and were found to be equal to 91 mK."

Again, I am missing some more information on how the non-blackness and the non-uniformity as well as the temperature uncertainty iteself have been estimated / determined and how these different sources of uncertainty contribute to the 91 mK.

This information has been reported from the characterization tests performed on the black body by the manufacturer.

**Line 145:**

"A possible instability of the temperature of different instrument components during the calibration process may affect the intensity of Earth View and Cold Space straylight."

Can you explain a bit more how temperature instabilities impact the straylight? Are the different instrument components themeselves the sources of straylight? Or does their temperature affect their behaviour of straying light from other sources? Please clarify.

This phrase refers to instrument components whose temperature, and therefore their emissivity, could potentially vary during the calibration process, thus affecting the straylight hitting the detector.

We have slightly rephrased the sentence in the revised version.

**Figure 7:**

Can you please indicate the temperature of the black body scenes in the figure caption?

We have modified the caption

Technical corrections / suggestions:
Line 104:
"The calculated coefficients, ..."
I suggest to write:
"The calculated coefficients for each pixel PN1 to PN4,
Line 105:
I suggest to delete "Therefore"
Line 204:
Typo: "at al." -> et al.
Line 206:

**"stemming by" -> stemming from**

The corrections have been made

**• Reviewer 3:**

This article by Dimitrios Kilymis et al. describing the uncertainty in the IASI radiances is both very important and comprehensive and well written, and somewhat overdue. The IASI radiances (combined with those from CrIS) are of great value for the primary purposes of medium range weather forecasting, thermodynamic profiling, and atmospheric composition, but also for establishing a record of hyperspectral infrared radiances which provide a wealth of information to characterize changes in the Earth's climate. The paper provides information on the various sources and size of uncertainty in the calibrated radiances, which is the information needed to interpret trends in the observed radiance record ... information which is much better and more useful than the overall sensor/contract spec of 0.5K at 280K. I suggest publishing as is.

**Response**

The authors would like to thank the reviewer for the positive feedback.

---

## Author Response (AR2)

Dear Dr Hoffmann,

We would like to thank you for accepting this manuscript for publication.

Regarding your request for higher quality files in the case of Figs. 11, 13 and 14, we are hereby submitting slightly better versions, taking into account that these are produced by operational software and therefore not modifiable. We hope that these files reach the journal standards for publication.

Best regards, Dimitrios Kilymis